# Effectiveness of Chitosan as a Dietary Supplement in Lowering Cholesterol in Murine Models: A Meta-Analysis

**DOI:** 10.3390/md19010026

**Published:** 2021-01-09

**Authors:** Sung-Il Ahn, Sangbuem Cho, Nag-Jin Choi

**Affiliations:** Department of Animal Science, Jeonbuk National University, Jeonju 54896, Korea; ahnsi71504@jbnu.ac.kr (S.-I.A.); chosb73@jbnu.ac.kr (S.C.)

**Keywords:** meta-analysis, chitosan, lifestyle-related disease, cholesterol lowering

## Abstract

This study presents a meta-analysis of studies that investigate the effectiveness of chitosan administration on lifestyle-related disease in murine models. A total of 34 published studies were used to evaluate the effect of chitosan supplementation. The effect sizes for various items after chitosan administration were evaluated using the standardized mean difference. Using Cochran’s Q test, the heterogeneity of effect sizes was assessed, after which a meta-ANOVA and -regression test was conducted to explain the heterogeneity of effect sizes using the mixed-effect model. Publication bias was performed using Egger’s linear regression test. Among the items evaluated, blood triglyceride and HDL-cholesterol showed the highest heterogeneity, respectively. Other than blood HDL-cholesterol, total cholesterol, and triglyceride in feces, most items evaluated showed a negative effect size with high significance in the fixed- and random-effect model (*p* < 0.0001). In the meta-ANOVA and -regression test, administering chitosan and resistant starch was revealed to be most effective in lowering body weight. In addition, chitosan supplementation proved to be an effective solution for serum TNF-α inhibition. In conclusion, chitosan has been shown to be somewhat useful in improving symptoms of lifestyle-related disease. Although there are some limitations in the results of this meta-analysis due to the limited number of animal experiments conducted, chitosan administration nevertheless shows promise in reducing the risk of cholesterol related metabolic disorder.

## 1. Introduction

Lifestyle-related diseases, including obesity, hyperlipidemia, atherosclerosis, type II diabetes, and hypertension, are widespread in industrialized countries, and are major threats to cardiovascular health. The syndrome is related to a combination of metabolic disorders, including abdominal obesity, hypertriglyceridemia, high-density lipoprotein (HDL) cholesterol decrease, hypertension, and high blood glucose, which lead to increased cardiovascular morbidity and mortality [1]. Unnatural blood lipid levels such as high levels of total cholesterol (TC) or triglyceride (TG), high low-density lipoprotein (LDL) level, or low HDL-cholesterol level are correlated with heart disease and stroke. Hypertension is one of the harmful risk factors for stroke and is a key factor in heart attacks. Moreover, obesity acts as a significant risk factor for cardiovascular disease and susceptibility to diabetes [2]. Thus, there has been an urgent need for effective methods of controlling these health-related parameters, including food additives.

Chitosan is one of the polymers containing acetyl glucosamine and glucosamine. It may be obtained by hydrolyzing and converting chitin with alkali from crabs, shrimps, insects, mushrooms, and the cell walls of microorganisms. Chitosan manufacture by deacetylation of chitin has been utilized in wastewater treatment and the agricultural sector. As the safety of chitin or chitosan has become increasingly recognized, it has recently-been used in a variety of fields, including medical supplies, food additives, and cosmetics [3,4]. Chitosan is also known among food additives of which the effects include lowering blood or liver cholesterol and triglyceride by combining with lipids [5]. It even shows an anti-inflammatory effect by TNF-α inhibition [6,7,8,9]. Nauss et al. [10] assume that chitosan binds lipid micelle in the small intestine after the ingestion of a fatty meal, while Kanauchi et al. [11] propose a more specific mechanism by which chitosan inhibits fat digestion in the gastrointestinal tract. In the stomach, chitosan is dissolved in acidic gastric juice. In this aqueous phase, it acts as an emulsifier on fat globules. It also mixes with fat to form an emulsion. Once transferred into the intestine, the chitosan in the emulsion turns into an insoluble gel-like form trapped fat, which cannot be decomposed by enzymes such as pancreatin or other intestinal enzymes. As a result, fat excretion in feces is increased (Figure 1). In this connection, [12] have confirmed that in one animal study chitosan administration led to fecal fat excretion approximately 7.5 times higher compared to that of a cellulose-fed group.

Meta-analysis is a method of statistical analysis that combines results from various scientific studies to obtain a quantified synthesis [13]. Meta-analysis increases the power of statistical analysis by pooling the results from multiple available studies. Therefore, this study summarizes the results of various animal experiments and provides integrated technical data for clinical trials so that clinical trials can proceed more accurately.

Studies of lifestyle diseases in murine models suggest that they may be improved by administering chitosan. However, few comprehensive studies have been conducted to date on the effect of chitosan supplementation on improving lifestyle diseases. Accordingly, the objective of the present study was to perform a meta-analysis of the effects of chitosan on factors in lifestyle-related diseases in adults.

## 2. Results

### 2.1. Data Set

Table 1 shows the data sets and experimental conditions for the 34 published studies used in the meta-analysis. The publication years of the studies ranged between 1978 and 2020. The animals most frequently used in the data set were rat strains such as Sprague-Dawley and Wistar, experiment duration was distributed between 2.8 and 21 weeks, and experimental diet most used for inducing hyperlipidemia in the data set was a high fat/cholesterol diet. In the case of Liu et al. [14], a high-fructose diet was used to induce hyperlipidemia. Furthermore, in the study of Gallaher et al. [12], blood total cholesterol (TC) was observed in all studies. In addition to total triglyceride (TG), low-density lipoprotein (LDL)- and high-density lipoprotein (HDL)-cholesterol in the blood, TC and TG in the liver, and fecal TC and TG were investigated. The levels of chitosan administered to hyperlipidemia-induced animals ranged from 0.045 to 7.5% of the diet. The chitosan administration period varied between 3 and 21 weeks.

### 2.2. Effect Size and Heterogeneity

The effect sizes of chitosan administration on hyperlipidemia in murine models using fixed and random effect models are listed in Table 2. Most items other than HDL-cholesterol in blood, total cholesterol, and triglyceride in feces showed negative effect size and high significance (*p* < 0.0001) in both effect models. These results mean that chitosan administration results in decreased levels of TC, TG, and LDL-C in blood, TC and TG in the liver, serum TNF-α and glucose in blood and body weight, and increased levels of blood HDL-C, fecal TC and TG.

### 2.3. Moderator Analysis

Since heterogeneity analysis in this study revealed a high level of heterogeneity between the studies analyzed, moderator analysis was performed to account for this. For this, meta-ANOVA and meta-regression were conducted. To perform the meta-ANOVA test, Q statistics between the subgroups (Q_b_) calculated under assessing that between subgroups (τ^2^) was the same. First of all, a meta-ANOVA analysis was performed on most items except fecal TG, as shown in Table 3 and Table 4. Chitosan and resistant starch (CTS + RS) showed the highest effect size in blood TC and TG, body weight, blood glucose and blood HDL-C, CTS showed the largest effect size in blood LDL-C and TNF-α, and the cholestyramine (CSR) and water-soluble chitosan (WSC) showed the greatest effect size in liver TC and liver TG, respectively (Table 3). However, none of these items were statistically significant (*p* < 0.05). Table 4 shows the results of meta-ANOVA in analyzing the effect of chitosans administration period on biological indices (*p* > 0.05). Other than fecal TC, body weight, and blood glucose, most items showed significant differences (*p* < 0.05). In the case of TC, the Q statistics between the groups (Q_b_) was 31.94 (df = 13, *p* = 0.0025); the effect size between groups was assumed to be significantly different.

Next, meta-regression was performed to evaluate the effect size between the type of chitosan used and the administration period (Table 5). Only CTS + RS was significant (*p* = 0.0208), and it was revealed to use to decrease blood TC. In the case of WSC, it was significantly effective in lowering of serum TNF-α and body weight (*p* = 0.0307 and 0.0008, respectively). With regard to the administration period, this was significantly relevant to blood HDL-C and liver TC with *p* = 0.0004 and 0.0358, respectively.

### 2.4. Publication Bias

Publication bias was conducted using an Egger’s linear regression test (Table 6) on blood TC and TG, blood LDL-C and HDL-C, liver TG and TC, fecal TC, and body weight. As the results from the Egger’s linear regression test show, significance was detected in all items (*p* < 0.05) indicating that the relationship between effect size and standard error was statistically significant and confirming the presence of bias [48]. Thus, the trim-and-fill technique was used to correct asymmetry due to publication bias in all items, with the resulting compensated effect sizes being shown in Table 7. Other than blood HDL-C, most of the effects showed significance (*p* < 0.05).

## 3. Discussion

In the results of Table 2, most items showed negative effect size and high significance (*p* < 0.0001) in both effect models. These results mean that chitosan administration results in decreased levels of TC, TG, and LDL-C in blood, TC and TG in the liver, TNF-α and glucose in blood and body weight, and increased levels of blood HDL-C, fecal TC and TG.

The bioavailability of dietary fat in the intestine decreased after chitosan administration. After this, reverse cholesterol transport, which is delivered from peripheral tissues to the liver, is accelerated by excretion of surplus dietary fat, resulting in an increase in the ratio of HDL-cholesterol [49]. Similarly, [50] have reported that the addition of chitosan to an animal diet caused a decrease in LDL-cholesterol content. Generally, HDL-cholesterol may decrease cardiovascular disease by converting cholesterol condensed on peripheral tissues or blood vessel walls into an ester compound. The ester compound is then transferred to the liver, excreted by bile-salt, and cholesterol content in blood is lowered. By contrast, LDL-cholesterol, which is the most general delivery type of blood cholesterol, accumulates easily on artery walls, causing arteriosclerosis. For this reason, it is known as the leading risk factor for arteriosclerosis and cardiovascular [51]. In this result, increased HDL-cholesterol, fecal total cholesterol, and triglyceride after chitosan administration are related to the factors mentioned above. According to Jeon and Kim [52], when chitosan is cationized (–NH_3_^+^), its viscosity is increased by the formation of poly cations and gels. In high viscosity of the intestine, dietary fiber lower blood cholesterol by delaying cholesterol diffusion from micelle to mucosa, inhibiting bile acid metabolism, delaying micelle forming, and reducing cholesterol absorption rate in the intestine [19,53]. Based on this result, chitosan exhibits an excellent anti-hypercholesterolemic effect and is thought to be effective in mitigating cardiovascular disease caused by excessive fat intake.

Cytokines are secreted by activated lymphocytes and macrophages, and regulate the function of the cells related to immune response. They are also recognized as playing an essential role in the inflammatory response [54]. Yemak et al. [8] report that TNF-α generation was lower in lipopolysaccharide (LPS) and chitosan-injected mice than in LPS-injected mice. Similarly, Seo et al. [7] observed that TNF-α was increased by the application of special stimulants in a human mast cell line (HMC−1), but decreased by the use of chitosan. TNF-α is one of the pro-inflammatory cytokines synthesized by adipose tissue [55,56], and high TNF-α levels are one of the critical risk factors for diabetes [57]. In a similar vein, Yoon et al. [58] state that chitosan is associated with an anti-inflammatory response to TNF-α gene expression. According to Zhu et al. [59], chitosan has an anti-inflammatory effect on active molecules, for example TNF-α and IL-1β via the NF-κB pathway. Activated macrophages secrete numerous pro-inflammatory cytokines, including IL-1β and TNF-α, to intermediate the inflammatory response [60]. However, overproduction of these pro-inflammatory mediators causes excessive inflammation [61]; thus, regulation of the release of pro-inflammatory mediators may be important in mitigating the inflammatory response.

According to Prabu and Naturajan [62], blood glucose levels decreased in streptozotocin-induced diabetic rats that were fed chitosan for 30 days. Other researchers suggest that the effectiveness of chitosan in lowering blood glucose may be due in part to the effect of total glyceride in lowering free fatty acids. Jo et al. [63] report that in an animal study, chitosan that was enzymatically treated and of low molecular weight (<1000 Da) was more effective in managing prandial glucose. Kim et al. [64] also report that chitosan that is low in molecular weight acted similarly to acarbose, a known anti-diabetic medication, in a murine model. They also note that chitosan administration inhibited sucrase and glucoamylase activities. It is recognized that chitosan binds with glucosidase in the intestinal brush border in a manner similar to acarbose (Hanefeld, [65]; Puls et al. [66]; Krentz and Bailey [67]). The inference of these reports is that body weight may be decreased by chitosan administration.

In the course of this process, heterogeneity is introduced as a result of methodological differences between studies. In general, a heterogeneity test is used to decide on methods for combining studies and to evaluate the consistency or inconsistency of findings (Petitti [68]; Higgins et al. [69]). To evaluate heterogeneity in relation to effect size in the present study, Q statistics and I^2^ values were computed. The highest among Q statistics was TG in blood, with high significance (*p* < 0.0001). The significance of the Q statistic implies that the studies used to calculate the overall effect (the effect size of fixed and random effect models) do not share the same effect size with one another (Cho et al. [70]). In this study, the Q statistics for all items were found to be significant (*p* < 0.0001). However, one limitation of this method is its dependence on the number of studies (Fleiss [71]). I^2^ and τ^2^ values are commonly used to overcome this limitation of Q statistics by providing a concrete indication of heterogeneity. The I^2^ value is used most frequently in meta-analysis to compare different numbers of studies and data types. Consequently, it offers a solution to the issue of the Q statistic when analyzing heterogeneity (Higgins et al. [72]). All items of I^2^ value in the present study were above 70%, which means that they all showed significant levels of heterogeneity [73]. The τ^2^ value indicates the absolute value of heterogeneity, representing variance in true effect sizes [74]. In addition, liver TG showed the highest τ^2^ value, which means that variance in the effectiveness of chitosan administration is great (Cho et al. [70]).

Cholestyramine (trade name: Questran, Questran Light, Cholybar or Olestyr) and cholestipol (trade name: Colestid or Cholestabyl) as an anion-exchanger are these days used mainly for reducing cholesterol [75]. These medications contain amino groups, are water-insoluble, and unlike chitosan are not absorbed in the intestine. Specifically, they form insoluble complexes with bile acids in the intestines, which are then excreted in the feces. As a result, more plasma cholesterol is converted into bile acids in the liver to normalize its levels. When cholesterol is converted into bile acids, plasma cholesterol levels are lowered (National Institute of Diabetes and Digestive and Kidney Diseases [76]). Consequently, they are known to inhibit cholesterol absorption in the gut and to promote bile salt excretion. However, they are also known to involve a number of issues, including gastrointestinal disturbance, constipation, and colon cancer [77,78]. Valhouny et al. [79] report that chitosan supplementation showed a similar inhibition effect to cholestyramine in cholesterol adsorption. Similarly, an animal study by Jennings et al. [78] showed that chitosan was similar to cholestyramine in lowering lipids without other harmful changes in intestinal mucosa. Currently, a total of 1832 patents related to chitosan are being searched in the field of hyperlipidemia and associated cardiovascular diseases. It can thus be concluded that chitosan supplementation may be useful in lowering cholesterol and offers a promising alternative treatment for lifestyle-related diseases.

## 4. Materials and Methods

### 4.1. Data Set

To perform a meta-analysis of published studies regarding the effect of chitosan administration on lowering cholesterol in murine models between 1978 to 2020, a literature search was conducted on Pubmed (US National Library of Medicine, Bethesda, MD, USA) and Science Direct (Elsevier B. V., Amsterdam, The Netherlands). The keywords used for searching studies for meta-analysis were “chitosan, cholesterol” in all databases. The results obtained included 450 citations from Science Direct and 303 from Pubmed (US National Library of Medicine, Bethesda, MD, USA). These results were then filtered by title, abstract, and full text. Among them, 4 review articles and 7 studies of clinical tests in human studies were removed. Also, the studies expressed with graphical data were eliminated. Following this, studies regarding changes in cholesterol levels after chitosan administration were collected. Ultimately, a total of 34 studies with 11 items (e.g., total cholesterol, triglyceride, LDL- and HDL-cholesterol, TNF-α, and so on) were selected to perform a meta-analysis of the effectiveness of chitosan in reducing cholesterol in murine models.

### 4.2. Data Analysis

Corrected standardized mean difference (Hedges’ g), and 95% confidence intervals (CI) were computed between control groups and treatment. The weight of the effect size was calculated using inverse-variance [80,81]. Effect-size analysis of fixed and random effect models was used to calculate overall effect due to differences in administration period, animal strain, and the type and dosage of chitosan used in each study. Cochran’s Q test was performed to assess the statistical heterogeneity of the effect size, and the ratio of true heterogeneity to total variation in observed effects was expressed by the I^2^ value. To confirm the heterogeneity of effect size using a mixed-effect model for the items in question, meta-ANOVA and regression analyses were also used. Meta-ANOVA and meta-regression analysis can evaluate the difference of Hedges’ g among subgroups herein administration periods or type of treatment. The periods were set as independent factors in meta-ANOVA and as continuous variables in meta-regression. Finally, publication bias analysis was conducted to ensure the validity of the meta-analysis results. Statistical analysis and visualization of the results were performed using the ‘meta’, and ‘metafor’ packages in the R statistics software application (ver. 3.5.3, R Foundation for Statistical Computing, Vienna, Austria).

## 5. Conclusions

The present study confirmed the effectiveness of chitosan administration on lifestyle-related diseases through meta-analysis. Chitosan was significantly effective in lowering total cholesterol and triglyceride of blood and liver and rising fecal total cholesterol and triglyceride. Based on our results, chitosan was demonstrated to be useful in improving the symptoms of lifestyle-related disease.

## Figures and Tables

**Figure 1 marinedrugs-19-00026-f001:**
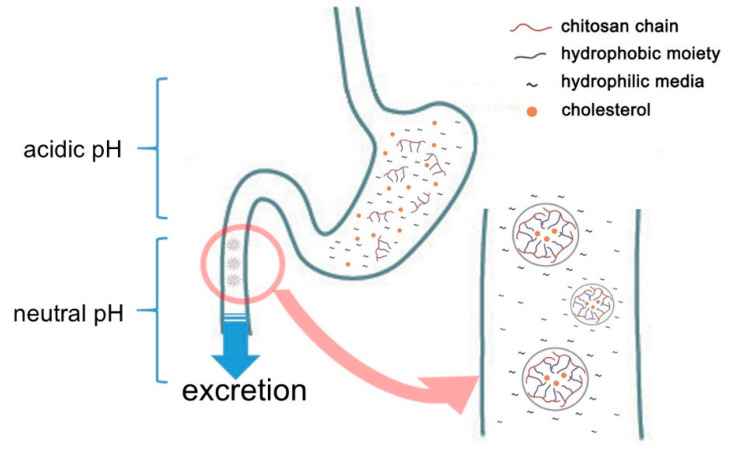
Schematic representation of cholesterol adsorption of chitosan in gastrointestinal tracts.

**Table 1 marinedrugs-19-00026-t001:** Studies used in the data set and their information for meta-analysis.

Authors	Animal (Strain)	n	Week	Experimental Diet	Analytical Items ^1^
Liu et al. (2018) [15]	Rat (Sprague–Dawley)	8	8	High fat	TC *, TG *, LDL-C *, HDL-C *, TNF-α *
Abozaid et al. (2015) [16]	Rat (white Albino)	10	6	High fat	TC *, TG *, LDL-C *, HDL-C *, TNF-α *
Bahijri et al. (2017) [17]	Rat (Wistar)	10	12	High fat	TC *, TG *, LDL-C *, HDL-C *
Chiu et al. (2015) [18]	Rat (Sprague–Dawley)	8	7	High fat	TC *, TG *, TC ^‡^, TG ^‡^
Park et al. (2010) [19]	Rat (Sprague–Dawley)	8	8	High fat	TC *, TG *, LDL-C *, HDL-C *, TC ^†^, TG ^†^, TC ^‡^
Sivakumar et al. (2007) [20]	Rat (Wistar)	6	8.5	High fat	TC *, TG *, LDL-C *, HDL-C *
Sugano et al. (1978) [21]	Rat (Wistar)	6	2.8	High fat	TC *, TG *, TC ^†^, TG ^†^, TC ^‡^
Tao et al. (2011) [22]	Rat (Sprague–Dawley)	8	4	High fat	TC *, TG *, LDL-C *, HDL-C *
Zacour et al. (1992) [23]	Rat (Wistar)	6	6	High fat	TC *, TG *, TC ^†^, TG ^†^, TC ^‡^, TG ^‡^
Yao and Chiang (2006) [24]	Hamster	9	8	High fat	TC *, TG *, LDL-C *, HDL-C *, TC ^†^, TG ^†^, TC ^‡^
Moon et al. (2007) [25]	Rat (Sprague–Dawley)	8	4	High fat	TC *, TG *, LDL-C *, HDL-C *, TC ^†^
Chiu et al. (2017) [26]	Rat (Sprague−Dawley)	8	5	High fat	TC *, TG*, HDL-C *, TC ^‡^, TG ^‡^
Liu et al. (2015) [14]	Rat (Sprague–Dawley)	8	21	High fructose	TC *, TG *, HDL-C *, TC ^†^, TG ^†^, TC ^‡^, TG ^‡^
Ardakani et al. (2009) [27]	Rat (Wistar)	5	2	High fat	TC *, TG *, LDL-C *, HDL-C *
Jung et al. (2016) [28]	Rat (Sprague–Dawley)	8	6	High fat	TC *, TG *, LDL-C *, HDL-C *
Hsieh et al. (2012) [29]	Rat (Sprague–Dawley)	9.5	10	High fat	TC ^†^, TG ^†^, TNF-α *
Han et al. (1999) [30]	Mouse (ICR)	13	9	High fat	TC *, TG, TC ^†^, TG ^†^, body weight
Chiang et al. (2000) [31]	Rat (Sprague–Dawley)	6	4	Normal diet + cellulose 5%	TC *, LDL-C *, HDL-C *, TC ^†^, TG ^†^, TC ^‡^, TG ^‡^
Shang et al. (2017) [32]	Rat (Sprague–Dawley)	8	6	High fat	TC *, TG *, LDL-C *, HDL-C *, body weight
Zhang et al. (2011) [33]	Rat (Sprague–Dawley)	8	4	High fat	TC *, TG *, LDL-C *, HDL-C *
van Bennekum et al. (2005) [34]	Mouse (C57BL/6)	6	3	High fat	TC *, TC ^†^
Zhou et al. (2008) [35]	Rat (Sprague–Dawley)	12	8	High fat	TC *, TG *, LDL-C *, HDL-C *, TNF-α *, glucose *
Kumar et al. (2009) [36]	Mouse (C57BL/6)	6	4	High fat	TC *, TG *
Kim et al. (2009) [37]	Rat (Sprague–Dawley)	5	8	High fat	TC *, body weight
Zong et al. (2012) [38]	Mouse (C57BL/6)	6	6	High fat	TC *, TG *, LDL-C *, HDL-C *, body weight,
Liu et al. (2012) [39]	Rat (Sprague–Dawley)	9	16	High sucrose	TC *, TG *, HDL-C *, TNF-α *, glucose *
Zhang et al. (2012) [40]	Rat (Sprague–Dawley)	8	8	High fat	TC *, TG *, LDL-C *, HDL-C *, TC ^†^, TG ^†^
Zhang et al. (2012) [41]	Rat (Sprague–Dawley)	10	4	High fat	TC *, TG *, LDL-C *, HDL-C *
Zhang and Xia (2015) [42]	Rat (Sprague–Dawley)	8	8	High fat	TC *, TG *, LDL-C *, HDL-C *, TC ^†^, TG ^†^, TC ^‡^, body weight
Si et al. (2017) [43]	Rat (Wistar)	8	6	High fat	TC *, TG *, LDL-C *, HDL-C *, body weight, glucose *
Do et al. (2018) [44]	Mouse (C57BL/6)	10	12	High fat	TC *, TG *, HDL-C *, TC ^†^, TG ^†^, TC ^‡^, TG ^‡^, body weight
Wang et al. (2019) [45]	Rat (Sprague–Dawley)	8	4.2	High fat	TC *, TG *, LDL-C *, HDL-C *, TC ^†^, TG ^†^, TC ^‡^, body weight
Chiu et al. (2020) [46]	Rat (Sprague–Dawley)	6	8	High fat	TC *, TC ^†^, TC ^‡^, TNF-α *
Wang et al. (2011) [47]	Rat (Wistar)	8	3	High fat	TG *, LDL-C *, HDL-C *

^1^ TC, total cholesterol; TG, triglyceride; LDL-C, low-density lipoprotein cholesterol; HDL-C, high-density lipoprotein cholesterol; TNF-α, tumor necrosis factor-α; *, blood; ^†^, liver; ^‡^, feces.

**Table 2 marinedrugs-19-00026-t002:** Effect size of chitosan administration on hyperlipidemia in murine model.

Items	df	Fixed Effect Model		Random Effect Model		Heterogeneity		
		ES ^1^	*p*-Value	ES	*p*-Value	Q (*p*-Value)	I^2^ (%)	τ^2^
Total cholesterol (blood)	65	−1.5457	<0.0001	−2.2248	<0.0001	376.43 (<0.0001)	82.7	2.1388
Triglyceride (blood)	63	−0.5852	<0.0001	−1.2366	<0.0001	525.93 (<0.0001)	88.0	2.6610
LDL-cholesterol (blood)	46	−1.6121	<0.0001	−2.5212	<0.0001	294.88 (<0.0001)	84.4	2.5182
HDL-cholesterol (blood)	49	0.1318	0.1363	0.1532	0.5704	431.89 (<0.0001)	88.7	3.0718
Total cholesterol (liver)	30	−2.3101	<0.0001	−3.3734	<0.0001	187.28 (<0.0001)	84.0	3.2403
Triglyceride (liver)	22	−2.1172	<0.0001	−3.2648	<0.0001	172.75 (<0.0001)	87.3	3.8731
Total cholesterol (feces)	22	1.8491	<0.0001	2.6038	<0.0001	113.25 (<0.0001)	80.6	2.2198
Triglyceride (feces)	9	2.0168	<0.0001	2.4130	<0.0001	35.30 (<0.0001)	74.5	1.5050
TNF-α (blood)	12	−1.4885	<0.0001	−1.8355	<0.0001	66.72 (<0.0001)	82.0	1.8174
Body weight	21	−1.5974	<0.0001	−2.4442	<0.0001	162.18 (<0.0001)	87.1	3.1836
Glucose (blood)	12	−0.7512	<0.0001	−0.8958	0.0096	61.64 (<0.0001)	80.5	1.2356

^1^ ES: effect size.

**Table 3 marinedrugs-19-00026-t003:** Meta-ANOVA analysis of effect of chitosan type on biological indices.

Biological Index ^1^	Analysis Item ^2^	*K* ^3^	Fixed Effect Model			Random Effect Model			Q ^6^	τ^2 7^	I^2 8^	Q_b_ ^9^	df ^10^	*p*
			SMD ^4^	95%-CI ^5^		SMD	95%-CI							
				Lower	Upper		Lower	Upper						
TC (blood)	CTS	42	−1.5720	−1.7639	−1.3801	−2.0640	−2.5645	−1.5635	194.29	2.2266	78.9	12.60	7	0.0826
	WSC	17	−1.5434	−1.9066	−1.1801	−2.7620	−3.6088	−1.9153	145.47	2.2266	89.0			
	RS	2	−1.7624	−2.6836	−0.8412	−2.3197	−4.6315	−0.0080	6.88	2.2266	85.5			
	CE	1	−2.1859	−3.7413	−0.6305	−2.1859	−5.4984	1.1266	0.00	-^11^	-			
	CTS + RS	1	−8.9998	−12.7228	−5.2769	−8.9998	−13.7341	−4.2655	0.00	-	-			
	WSC + RS	1	−1.4835	−4.6243	1.6572	−1.4835	−4.6243	1.6572	0.00	-	-			
	CTS + VitC	1	0.2823	−0.7041	1.2688	0.2823	−2.8042	3.3688	0.00	-	-			
	CSR	1	−1.9182	−3.3870	−0.4494	−1.9182	−5.1910	1.3545	0.00	-	-			
TG (blood)	CTS	39	−0.4142	−0.6035	−0.2249	−1.0874	−1.6614	−0.5135	378.70	2.8583	90.0	5.50	6	0.4819
	WSC	17	−1.1778	−1.4773	−0.8782	−1.9030	−2.7835	−1.0224	86.48	2.8583	81.5			
	RS	2	−0.8491	−1.6483	−0.0499	−1.1971	−3.6866	1.2924	7.64	2.8583	86.9			
	CE	1	1.1106	0.0996	2.1216	1.1106	−2.3538	4.5750	0.00	-	-			
	CTS + RS	1	−3.4066	−5.0825	−1.7307	−3.4066	−7.1199	0.3067	0.00	-	-			
	WSC + RS	1	−1.4338	−2.5685	−0.2990	−1.4338	−4.9363	2.0688	0.00	-	-			
	CTS + VitC	1	−0.8606	−1.8990	0.1778	−0.8606	−4.3331	2.6119	0.00	-	-			
LDL-C (blood)	CTS	28	−2.1800	−2.4471	−1.9129	−2.8041	−3.4238	−2.1843	100.39	1.9848	73.1	6.27	3	0.0990
	WSC	13	−1.6760	−2.0564	−1.2956	−2.8831	−3.8113	−1.9550	100.21	1.9848	88.0			
	RS	2	−0.2492	−0.9457	0.4474	−0.2492	−2.3222	1.8238	0.00	1.9848	0.0			
	CTS + RS	2	−1.6799	−2.5327	−0.8270	−1.7721	−3.9089	0.3647	1.49	1.9848	32.9			
HDL-C (blood)	CTS	36	0.1332	−0.0704	0.3368	0.3816	−0.2696	1.0329	315.19	3.3880	88.9	3.63	4	0.4585
	WSC	10	−0.0158	−0.4449	0.4132	−0.7968	−2.0465	0.4528	107.00	3.3880	91.6			
	RS	2	−0.1120	−0.8081	0.5842	−0.1134	−2.7577	2.5308	0.34	3.3880	0.0			
	CTS + RS	1	1.9999	0.7360	3.2638	1.9999	−1.8227	5.8225	0.00	-	-			
	WSC + RS	1	0.2293	−0.7549	1.2135	0.2293	−3.5102	3.9688	0.00	-	-			
TC (liver)	CTS	26	−2.5523	−2.8603	−2.2442	−3.6571	−4.4528	−2.8614	157.12	3.2800	84.1	4.50	3	0.2122
	WSC	3	−1.0529	−1.7573	−0.3485	−1.5068	−3.6972	0.6837	11.05	3.2800	81.9			
	CE	1	−1.5873	−2.9588	−0.2158	−1.5873	−5.3927	2.2181	0.00	-	-			
	CSR	1	−4.7470	−7.3259	−2.1682	−4.7470	−9.1346	−0.3595	0.00	-	-			
TG (liver)	CTS	19	−1.9028	−2.2234	−1.5823	−3.0600	−4.0045	−2.1154	153.33	3.5904	88.3	0.91	1	0.3410
	WSC	4	−3.9955	−4.9445	−3.0466	−4.1792	−6.2803	−2.0781	2.65	3.5904	0.0			
TC (feces)	CTS	20	1.8847	1.5660	2.2034	2.6479	1.8759	3.4200	97.07	2.3783	80.4	0.04	1	0.8341
	WSC	3	1.6188	0.8072	2.4304	2.4194	0.4258	4.4131	15.82	2.3783	87.4			
Body weight	CTS	11	−2.4795	−2.9132	−2.0458	−3.4586	−4.5418	−2.3755	78.81	2.5667	87.3	18.75	4	0.0009
	WSC	7	−0.5100	−0.9616	−0.0584	−0.5950	−1.8669	0.6769	22.98	2.5667	73.9			
	RS	2	−1.7624	−2.6836	−0.8412	−2.3356	−4.7858	0.1147	6.88	2.5667	85.5			
	CTS + RS	1	−8.9998	−12.7228	−5.2769	−8.9998	−13.8702	−4.1295	0.00	-	-			
	WSC + RS	1	−1.4835	−2.6285	−0.3386	−1.4835	−4.8258	1.8588	0.00	-	-			
TNF-α	CTS	12	−1.6953	−2.0508	−1.3398	−2.0430	−2.8184	−1.2676	49.88	1.4116	77.9	19.84	3	0.0002
	WSC	1	0.9843	−0.2451	2.2137	0.9843	−1.6489	3.6175	0.00	-	-			
Glucose (blood)	CTS	10	−0.7573	−1.0898	−0.4247	−0.9044	−1.6869	−0.1218	48.68	1.2809	81.5	2.49	3	0.4765
	RS	1	−1.6688	−2.8537	−0.4840	−1.6688	−4.1837	0.8460	0.00	-	-			
	CTS + RS	1	−1.7693	−2.9772	−0.5615	−1.7693	−4.2951	0.7564	0.00	-	-			
	CTS + VitC	1	0.7144	−0.3062	1.7350	0.7144	−1.7274	3.1562	0.00	-	-			

^1^ TC, total cholesterol; TG, triglyceride, LDL-C, low-density lipoprotein; HDL-C, high-density lipoprotein, TNF-α; Tumor necrosis factor alpha; ^2^ CTS, chitosan; WSC, water-soluble chitosan; RS, resistant starch; CE, cellulose; CTS + RS, chitosan and resistant starch; WSC + RS, water-soluble chitosan and resistant starch; CTS + VitC, chitosan and vitamin C; CSR, cholestyramine; ^3^
*k*: number of treatments; ^4^ SMD: standardized mean difference; ^5^ CI: confidence interval; ^6^ Q: chi-squared statistic; ^7^ τ^2^: true heterogeneity; ^8^ I^2^: Higgin’s I^2^ statistic; ^9^ Q_b_: Q statistics between groups; ^10^ df: degrees of freedom of Q statistic; ^11^ –: no data.

**Table 4 marinedrugs-19-00026-t004:** Meta-ANOVA analysis of effect of chitosans administration period on biological indices.

Item ^1^	Administration Period (Week)	*K* ^2^	Fixed Effect Model			Random Effect Model			Q ^5^	τ^2 6^	I^2 7^	Q_b_ ^8^	Df ^9^	*p*
			SMD ^3^	95%-CI ^4^		SMD	95%-CI							
				Lower	Upper		Lower	Upper						
TC (blood)	2	1	−2.4519	−4.3072	−0.5966	−2.4519	−5.6070	0.7668	0.00	-^10^	-	31.94	13	0.0025
	2.8	1	−5.4530	−8.3456	−2.5605	−5.4530	−9.3626	−1.5435	0.00	-	-			
	3	3	−1.7388	−2.5626	−0.9149	−1.7849	3.5161	−0.0536	0.84	1.8009	0.0			
	4	15	−2.4145	−2.7809	−2.0482	3.0386	−3.8429	−2.2343	58.25	1.8009	76.0			
	4.2	4	−1.4345	−2.0713	−0.7978	−2.1531	−3.6612	−0.6451	19.86	1.8009	84.9			
	5	1	−2.1630	−3.4689	−0.8570	−2.1630	−5.0996	0.7736	0.00	-	-			
	6	13	−0.8035	−1.1424	−0.4645	−1.3554	−2.1863	−0.5246	73.20	1.8009	83.6			
	7	3	0.1396	−0.4290	0.7082	0.1408	−1.4807	1.7624	0.48	1.8009	0.0			
	8	13	−1.9919	−2.3823	−1.6016	−2.4673	−3.3262	−1.6083	45.48	1.8009	73.6			
	8.5	1	−2.6715	−4.3982	−0.9449	−2.6715	−5.8178	0.4748	0.00	-	-			
	9	3	−3.9353	−4.8325	−3.0381	−5.3277	−7.2274	−3.4280	26.07	1.8009	92.3			
	12	4	−1.1933	−1.8028	−0.5839	−1.8238	−3.3133	−0.3343	31.08	1.8009	90.3			
	16	3	−1.1877	−1.7836	−0.5919	−1.2240	−2.8565	0.4085	1.48	1.8009	0.0			
	21	1	−1.2794	−2.3845	−0.1744	−1.2794	−4.1324	1.5735	0.00	-	-			
TG (blood)	2	1	−1.7652	−3.3540	−0.1764	−1.7652	−4.6864	1.1560	0.00	-	-	96.55	13	<0.0001
	2.8	1	1.5689	0.2025	2.9353	1.5689	−1.2375	4.3754	0.00	-	-			
	3	2	−1.1789	−1.9477	−0.4101	−1.1794	−3.0756	0.7168	0.01	1.5643	0.0			
	4	15	−0.6150	−0.9114	−0.3186	−1.1423	−1.8601	−0.4246	115.66	1.5643	87.9			
	4.2	4	−3.0564	−3.8949	−2.2179	−3.6529	−5.1879	−2.1179	12.50	1.5643	76.0			
	5	1	−0.5418	−1.5454	0.4617	−0.5418	−3.1906	2.1070	0.00	-	-			
	6	13	−0.9085	−1.2282	−0.5889	−1.0837	−1.8397	−0.3276	38.28	1.5643	68.6			
	7	3	1.1330	0.5098	1.7563	1.1336	−0.4129	2.6800	0.02	1.5643	0.0			
	8	10	−0.8028	−1.1722	−0.4333	−1.3193	−2.1969	−0.4417	68.52	1.5643	86.9			
	8.5	1	−2.5453	−4.2258	−0.8647	−2.5453	−5.5173	0.4268	0.00	-	-			
	9	3	−9.3202	−10.9958	−7.6445	−9.5824	−11.8082	−7.3566	4.23	1.5643	52.7			
	12	3	−1.6964	−2.3414	−1.0515	−2.0228	−3.5934	−0.4521	10.45	1.5643	80.9			
	16	4	1.3087	0.7842	1.8333	1.3374	0.0031	2.6716	1.33	1.5643	0.0			
	21	1	1.3630	0.2421	2.4838	1.3630	−1.3324	4.0584	0.00	-	-			
LDL-C (blood)	2	1	−1.7162	−3.2878	−0.1445	−1.7162	−4.3664	0.9341	0.00	-	-	59.48	7	<0.0001
	3	2	−0.2574	−0.9554	0.4407	−0.2590	−1.9216	1.4036	0.18	1.1854	0.0			
	4	15	−2.2993	−2.6446	−1.9539	−2.5700	−3.2315	−1.9085	33.05	1.1854	57.6			
	4.2	4	−11.5502	−13.9253	−9.1751	−11.6010	−14.2147	−8.9872	1.53	1.1854	0.0			
	6	11	−1.4658	−1.8446	−1.0870	−1.7983	−2.5618	−1.0348	32.11	1.1854	68.9			
	8	10	−1.7590	−2.1810	−1.3371	−2.5968	−3.4802	−1.7134	60.61	1.1854	85.2			
	8.5	1	−5.2140	−7.9995	−2.4285	−5.2140	−8.7229	−1.7050	0.00	-	-			
	12	1	−2.6102	−3.8684	−1.3520	−2.6102	−5.0875	−0.1329	0.00	-	-			
HDL-C (blood)	2	1	1.0239	−0.3431	2.3910	1.0239	−2.2340	4.2819	0.00	-	-	69.79	10	<0.0001
	3	2	0.5630	−0.6456	1.7715	0.5824	−1.8350	2.9998	0.18	2.2766	0.0			
	4	13	−2.0684	−2.4694	−1.6673	−2.9457	−3.883.7	−2.0077	106.17	2.2766	88.7			
	4.2	4	0.0038	−0.4905	0.4981	0.0015	−1.5576	1.5606	1.69	2.2766	0.0			
	5	1	0.4126	−0.5811	1.4064	0.4126	−2.7071	3.5324	0.00	-	-			
	6	13	0.5061	0.1888	0.8233	0.9245	0.0276	1.8213	63.04	2.2766	81.0			
	8	8	1.3101	0.9108	1.7093	1.5747	0.4469	2.7025	21.96	2.2766	68.1			
	8.5	1	5.6757	2.6829	8.6686	5.6757	1.4683	9.8832	0.00	-	-			
	12	3	0.2234	−0.6859	1.1328	4.0812	1.8229	6.3396	66.29	2.2766	97.0			
	16	3	0.4824	−0.1859	1.1506	0.4850	−1.3486	2.3185	0.14	2.2766	0.0			
	21	1	0.6661	−0.3493	1.6815	0.6661	−2.4607	3.7928	0.00	-	-			
TC (liver)	2.8	1	−9.3712	−14.0837	−4.6586	−9.3712	−14.5605	−4.1818	0.00	-	-	62.17	9	<0.0001
	3	3	−2.2552	−3.2123	−1.2980	−2.5488	−4.1731	−0.9244	4.50	1.2291	55.6			
	4	4	−1.1782	−1.8149	−0.5415	−1.6315	−2.9484	−0.3146	11.98	1.2291	75.0			
	4.2	4	−1.4115	−2.0064	−0.8165	−1.6940	−2.9526	−0.4353	9.42	1.2291	68.1			
	6	1	−1.6407	−3.0270	−0.2543	−1.6407	−4.2182	0.9368	0.00	-	-			
	8	10	−2.4940	−2.9903	−1.9978	−2.9111	−3.7901	−2.0320	32.45	1.2291	72.3			
	9	3	−9.3634	−11.0438	−7.6830	−9.5728	−11.6968	−7.4488	3.86	1.2291	48.2			
	10	2	−2.7055	−3.6429	−1.7681	−2.7400	−4.5432	−0.9368	0.52	1.2291	0.0			
	12	2	−6.7401	−8.5025	−4.9776	−6.7503	−9.0902	−4.4104	0.11	1.2291	0.0			
	21	1	−3.0412	−4.6012	−1.4813	−3.0412	−5.7161	−0.3664	0.00	-	-			
TG (liver)	2.8	1	−5.1384	−7.8902	−2.3866	−5.1384	−9.5841	−0.6927	0.00	-	-	18.28	8	0.0192
	4	2	0.6597	−0.1879	1.5072	0.7060	−1.9059	3.3180	1.18	3.1738	15.0			
	4.2	4	−2.8029	−3.5742	−2.0316	−3.1371	−5.0651	−1.2092	6.82	3.1738	56.0			
	6	1	−3.2107	−5.1438	−1.2775	−3.2107	−7.2018	0.7804	0.00	-	-			
	8	7	−2.5754	−3.2000	−1.9508	−4.0563	−5.5980	−2.5146	47.21	3.1738	87.3			
	9	3	−3.3928	−4.2013	−2.5842	−4.7613	−7.0104	−2.5123	24.84	3.1738	91.9			
	10	2	−0.9960	−1.6800	−0.3120	−0.9970	−3.5590	1.5650	0.03	3.1738	0.0			
	12	2	−5.0743	−6.5083	−3.6403	−5.6019	−8.5208	−2.6830	4.14	3.1738	75.8			
	21	1	−1.8937	−3.1313	−0.6562	−1.8937	−5.5982	1.8108	0.00	-	-			
TC (feces)	2.8	1	5.3232	2.4889	8.1575	5.3232	1.3385	9.3079	0.00	-	-	10.86	8	0.2098
	4	2	0.5976	−0.2454	1.4405	0.6403	−1.5136	2.7943	1.26	2.0420	20.6			
	4.2	4	1.6887	0.9671	2.4103	3.0580	1.3721	4.7439	30.95	2.0420	90.3			
	5	1	1.0557	−0.0110	2.1224	1.0557	−1.9414	4.0527	0.00	-	-			
	6	1	1.7250	0.3145	3.1356	1.7250	−1.4109	4.8609	0.00	-	-			
	7	3	3.0554	2.1511	3.9596	3.0669	1.2132	4.9206	0.18	2.0420	0.0			
	8	8	1.5481	1.0714	2.0247	2.2101	1.0727	3.3475	33.38	2.0420	79.0			
	12	2	4.1314	2.9406	5.3221	4.1879	1.8709	6.5048	0.59	2.0420	0.0			
	21	1	5.0436	2.8049	7.2823	5.0436	1.4581	8.6292	0.00	-	-			
TG (feces)	4	2	2.0809	1.0028	3.1590	2.0809	1.0028	3.1590	0.17	0.0000	0.0	34.97	5	<0.0001
	5	1	0.1343	−0.8472	1.1157	0.1343	−0.8472	1.1157	0.00	-	-			
	6	1	2.3328	0.7274	3.9382	2.3328	0.7274	3.9382	0.00	-	-			
	7	3	1.9419	1.2198	2.6640	1.9419	1.2198	2.6640	0.16	0.0000	0.0			
	12	2	5.2475	3.8224	6.6726	5.2475	3.8224	6.6726	0.00	0.0000	0.0			
	21	1	2.7213	1.2585	4.1841	2.7213	1.2585	4.1841	0.00	-	-			
Body weight	4.2	4	−1.7969	−2.4141	−1.1796	−1.8690	−3.7561	−0.0360	3.13	3.1917	4.1	8.74	4	0.0679
	6	9	−0.9479	−1.3802	−0.5155	−1.8184	−3.1069	−0.5299	56.45	3.1917	85.8			
	8	3	−1.4852	−2.2513	−0.7191	−1.8393	−4.0223	0.3436	9.29	3.1917	78.5			
	9	3	−3.9353	−4.8325	−3.0381	−5.6255	−7.9722	−3.2789	26.07	3.1917	92.3			
	12	3	−1.7793	−2.6526	−0.9059	−2.8871	−5.1625	−0.6117	31.84	3.1917	93.7			
TNF-α	6	1	−8.0454	−10.9625	−5.1282	−8.0454	−11.4788	−4.6119	0.00	-	-	19.84	3	0.0002
	8	7	−1.0126	−1.4568	−0.5683	−1.0557	−1.8732	−0.2382	27.60	0.8535	78.3			
	10	2	−3.4666	−4.5486	−2.3845	−3.4672	−5.1437	−1.7908	0.01	0.8535	0.0			
	16	3	−1.4696	−2.0976	−0.8416	−1.5301	−2.7528	−0.3075	2.46	0.8535	18.6			
Glucose (blood)	6	5	−0.4178	−0.9320	0.0963	−0.4410	−1.5333	0.6513	30.43	1.2036	86.9	8.51	5	0.1304
	8	1	−4.5622	−6.1849	−2.9395	−4.5622	−7.2560	−1.8684	0.00	-	-			
	10	2	−1.2618	−1.9991	−0.5246	−1.2927	−2.9841	0.3987	0.83	1.2036	0.0			
	12	1	−0.1355	−1.0133	0.7423	−0.1355	−2.4580	2.1870	0.00	-	-			
	16	3	−0.6143	−1.1747	−0.0522	−0.6727	−2.0368	0.6914	3.57	1.2036	43.9			
	21	1	−0.8705	−1.9103	0.1692	−0.8705	−3.2590	1.5179	0.00	-	-			

^1^ TC, total cholesterol; TG, triglyceride, LDL-C, low-density lipoprotein; HDL-C, high-density lipoprotein, TNF-α; Tumor necrosis factor alpha; ^2^
*k*: number of treatments; ^3^ SMD: standardized mean difference; ^4^ CI: confidence interval; ^5^ Q: chi-squared statistic; ^6^ τ^2^: true heterogeneity; ^7^ I^2^: Higgin’s I^2^ statistic; ^8^ Q_b_: Q statistics between groups; ^9^ df: degrees of freedom of Q statistic; ^10^ –: no data.

**Table 5 marinedrugs-19-00026-t005:** Meta-regression analysis of effect of chitosan on lowering cholesterol.

Item	Item ^1^		Estimate	SE	*p*-Value ^2^	ci. lb	ci. ub
TC (blood)	Type	Intercept	−2.1859	1.6901	0.1959	−5.4884	1.1266
		CTS	0.1219	1.7093	0.9431	−3.2282	3.4720
		WSC	−0.5761	1.7444	0.7412	−3.9952	2.8429
		RS	−0.1338	2.0610	0.9482	−4.1733	3.9056
		CTS + RS	−6.8139	2.9481	0.0208 *	−12.5920	−1.0358
		WSC + RS	0.7024	2.3290	0.7630	−3.8624	5.2671
		CSR	0.2677	2.3758	0.9103	−4.3889	4.9242
	Administ-ration period	Intercept	−2.7155	0.4412	<0.0001 ***	−3.5802	−1.8509
		Period	0.0701	0.0561	1.2503	−0.0398	0.1800
TG (blood)	Type	Intercept	1.1106	1.7676	0.5298	−2.3538	4.5750
		CTS	−2.1980	1.7917	0.2199	−5.7097	1.3136
		WSC	−3.0136	1.8238	0.0985	−6.5881	0.5610
		RS	−2.3077	2.1766	0.2890	−6.5738	1.9584
		CTS + RS	−4.5172	2.5911	0.0813	−9.5957	0.5613
		WSC + RS	−2.5444	2.5135	0.3114	−7.4708	2.3821
		CTS + VitC	−1.9712	2.5027	0.4309	−6.8764	2.9340
	Administ-ration period	Intercept	−2.0619	0.4644	<0.0001 ***	−2.9721	−1.1516
		Period	0.1108	0.0586	0.0586	−0.0040	0.2257
LDL-C (blood)	Type	Intercept	−1.7721	1.0902	0.1041	−3.9089	0.3647
		CTS	−1.0320	1.1352	0.1041	−3.9089	0.3647
		WSC	−1.1110	1.1886	0.3499	−3.4407	1.2186
		RS	1.5229	1.5190	0.3161	−1.4542	4.5000
	Administ-ration period	Intercept	−2.3459	0.7447	0.0016 **	−3.8056	−0.8863
		Period	−0.0554	0.1258	0.6595	−0.3021	0.1912
HDL-C (blood)	Type	Intercept	0.3816	0.3323	0.2507	−0.2696	1.0329
		WSC	−1.1785	0.7190	0.1012	−2.5877	0.2307
		RS	−0.4951	1.3894	0.7216	−3.2183	2.2282
		CTS + RS	1.6183	1.9784	0.4134	−2.2594	5.4959
		WSC + RS	−0.1523	1.9366	0.9373	−3.9481	3.6434
	Administ-ration period	Intercept	−1.4886	0.5323	0.0052 **	−2.5319	−0.4453
		Period	0.2432	0.0684	0.0004 ***	0.1091	0.3773
TC(liver)	Type	Intercept	−1.5873	1.9416	0.4136	−5.3927	2.2181
		WSC	0.0805	2.2402	0.9713	−4.3102	4.4713
		CTS	−2.0698	1.9835	0.2967	−5.9575	1.8179
		CSR	−3.1597	2.9633	0.2869	−8.9676	2.6481
	Administ-ration period	Intercept	−1.9173	0.7594	0.0116	−3.4057	−0.4289
		Period	−0.1982	0.0944	0.0358	−0.3872	−0.0132
TG(blood)	Type	Intercept	−3.0600	0.4819	<0.0001 ***	−4.0045	−2.1154
		WSC	−1.1192	1.1754	0.3410	−3.4229	1.1845
	Administ-ration period	Intercept	−2.7837	1.0596	0.0086 **	−4.8606	−0.7068
		Period	−0.0620	0.1197	0.6045	−0.2967	0.1727
TC (feces)	Type	Intercept	2.6479	0.3939	<0.0001 ***	1.8759	3.4200
		WSC	−0.2285	1.0908	0.8341	−2.3664	1.9094
	Administ-ration period	Intercept	1.3488	0.7729	0.0810	−0.1661	2.8637
		Period	0.1637	0.0958	0.0808	−0.0205	0.3552
TG (feces)	Administ-ration period	Intercept	1.2205	0.8155	0.1345	−0.3778	2.8189
		Period	0.1409	0.0847	0.0961	−0.2510	0.3069
TNF-α	Type	Intercept	−2.0430	0.3956	<0.0001 ***	−2.8184	−1.2676
		WSC	3.0273	1.4005	0.0307 *	0.2823	5.7723
	Administ-ration period	Intercept	−2.4611	1.3793	0.0744	−5.1646	0.2423
		Period	0.0599	0.1285	0.6413	−0.1920	0.3117
Body weight	Type	Intercept	−3.4586	0.5526	<0.0001 ***	−4.5418	−2.3755
		WSC	2.8636	0.8524	0.0008 ***	1.1930	4.5342
		RS	1.1231	1.3669	0.4113	−1.5559	3.8021
		CTS + RS	−5.5412	2.5456	0.0295 *	−10.5305	−0.5519
		WSC + RS	1.9751	1.7926	0.2705	−1.5383	5.4885
	Administ-ration period	Intercept	−0.5489	1.3274	0.6793	−3.1506	2.0529
		Period	−0.2678	0.1770	0.1303	−0.6148	0.0792
Glucose (blood)	Type	Intercept	−0.9044	0.3993	0.0235 *	−1.6869	−0.1218
		RS	−0.7644	1.3438	0.5694	−3.3982	1.8694
		CTS + RS	−0.8650	1.3491	0.5214	−3.5092	1.7793
		CTS + VitC	1.6188	1.3082	0.2159	−0.9453	4.1829
	Administ-ration period	Intercept	−1.0118	0.8754	0.2477	−2.7275	0.7039
		Period	0.0103	0.0736	0.8887	−0.1339	0.1545

^1^ CTS, chitosan; WSC, water-soluble chitosan; RS, resistant starch; CTS + RS, chitosan and resistant starch; WSC + RS, water-soluble chitosan and resistant starch; CTS + VitC, chitosan and vitamin C; CSR, cholestyramine; ^2^ Means marked with *, **, and *** differ significantly (*p* < 0.05, 0.01 and 0.001, respectively).

**Table 6 marinedrugs-19-00026-t006:** Egger’s linear regression test for publication bias.

Items	Bias	Se ^1^. bias	Slope	t	df ^2^	*p*-Value
Total cholesterol (blood)	−6.9521793	0.5168551	2.8826324	−13.451	64	<2.2 × 10^−16^
Triglyceride (blood)	−7.4780606	0.9998057	3.7716108	−7.4795	67	2.087 × 10^−10^
LDL-cholesterol (blood)	−6.1250126	0.4715822	2.2442145	−12.988	46	<2.2 × 10^−16^
HDL-cholesterol (blood)	0.51543585	1.43323094	−0.07605097	0.35963	52	<0.0001
Total cholesterol (liver)	−6.5468325	0.5461543	2.4287577	−11.987	30	5.732 × 10^−13^
Triglyceride (liver)	−6.7370699	0.9014982	2.5785977	−7.4732	21	2.411 × 10^−07^
Total cholesterol (feces)	6.5339622	0.4235035	−2.6905774	15.428	24	5.871 × 10^−14^
Triglyceride (feces)	8.411555	1.070048	−3.8220945	7.8609	8	4.953 × 10^−05^
TNF-α (blood)	−8.347186	2.266406	3.647681	−3.683	11	0.003607
Body weight	−7.798456	1.192187	3.513530	−6.5413	20	2.249 × 10^−06^

^1^ Se: standard error; ^2^ df: degrees of freedom of Q statistic.

**Table 7 marinedrugs-19-00026-t007:** Trimmed effect size of probiotics on inflammatory bowel disease in murine model.

Items	df	Fixed Effect Model		Random Effect Model		Heterogeneity		
		ES	*p*-Value	ES	*p*-Value	Q (*p*-Value)	I^2^ (%)	τ^2^
Total cholesterol (blood)	86	−1.1096	<0.0001	−1.2079	<0.0001	686.36 (<0.0001)	87.5	3.6291
Triglyceride (blood)	78	−0.2142	0.0029	−0.2935	0.2360	878.84 (<0.0001)	91.1	4.2254
LDL-cholesterol (blood)	64	−1.1291	<0.0001	−1.2373	<0.0001	551.97 (<0.0001)	88.4	4.2350
HDL-cholesterol (blood)	52	0.0607	0.4912	−0.1870	0.5174	521.81 (<0.0001)	90.0	3.7407
Total cholesterol (liver)	42	−1.7190	<0.0001	−1.8509	<0.0001	367.01 (<0.0001)	88.6	5.8208
Triglyceride (liver)	31	−1.4703	<0.0001	−1.6805	0.0004	314.64 (<0.0001)	90.1	6.2275
Total cholesterol (feces)	31	1.2437	<0.0001	1.3796	0.0004	226.78 (<0.0001)	86.3	3.9640
Triglyceride (feces)	13	1.4815	<0.0001	1.5692	0.0011	71.66 (<0.0001)	81.9	2.5666
TNF-α (blood)	14	−1.2869	<0.0001	−1.3743	0.0026	96.71 (<0.0001)	85.5	2.5645
Body weight	27	−1.1740	<0.0001	−1.2547	0.0079	284.88 (<0.0001)	90.5	5.3068
Glucose (blood)	12	−0.7512	<0.0001	−0.8958	0.0096	61.64 (<0.0001)	80.5	1.2356

## Data Availability

The data presented in this study are fully available in the main text of this article.

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
