# Peer review of "Effectiveness of Chitosan as a Dietary Supplement in Lowering Cholesterol in Murine Models: A Meta-Analysis"

_marinedrugs, 2021, doi:10.3390/md19010026_

Round 1

Reviewer 1 Report

This is an interesting meta-analysis concerning the potential ability to use chitosan to lower cholesterol.

The aim of the study is clear and the data are convincing.

However, the discussion needs to be improved.

In the discussion, could you give information related to the number of patents describing chitosan in the field of hypercholesterolemia and associated cardiovascular diseases?

Have there been any clinical studies with chitosan and are there any clinical studies underway?

Is it possible to hope for drugs with chitosan and, if so, which drugs would chitosan compete with?

Author Response

Reviewer 1

This is an interesting meta-analysis concerning the potential ability to use chitosan to lower cholesterol.

The aim of the study is clear and the data are convincing.

However, the discussion needs to be improved.

In the discussion, could you give information related to the number of patents describing chitosan in the field of hypercholesterolemia and associated cardiovascular diseases?

--> Attached information of the patents in the lines 96-97.

Have there been any clinical studies with chitosan and are there any clinical studies underway?

--> We know that a number of clinical studies have been conducted, however this is not mentioned here. Currently, we do not know about ongoing clinical studies.

Is it possible to hope for drugs with chitosan and, if so, which drugs would chitosan compete with?

--> Already decribed it in Line 83 in page 21. However, we supplemented more information. Please see lines 86-90 in page 21.

Please see the revised manuscript.

Reviewer 2 Report

In this work the authors present a meta-analysis of the effectiveness of chitosan ad dietary supplement in lowering cholesterol in murine models. 

The work can be interesting for researchers that are strongly specialized in the field, but it is difficult to follow for a broader audience. Conceptually it is a review, but the manuscript is perceived as an original work. Thefore deep re-structuring of the text body is suggested. 

Most of the text included in the Discussion should be moved to the Introduction. The Results should be joined with the Discussion and a clear Conclusion should be provided. The reader should be able to immediately extract the added-value of this manuscript for instance by reading a Table that summarizes the results of the work that is not present in the current manuscript. 

Author Response

Reviewer 2

In this work the authors present a meta-analysis of the effectiveness of chitosan ad dietary supplement in lowering cholesterol in murine models.

The work can be interesting for researchers that are strongly specialized in the field, but it is difficult to follow for a broader audience. Conceptually it is a review, but the manuscript is perceived as an original work. Thefore deep re-structuring of the text body is suggested.

Most of the text included in the Discussion should be moved to the Introduction. The Results should be joined with the Discussion and a clear Conclusion should be provided.

--> Some parts of Discussion moved to Introduction and Conclusion was described again clearly.

The reader should be able to immediately extract the added-value of this manuscript for instance by reading a Table that summarizes the results of the work that is not present in the current manuscript.

Thank you for your advice. Please see the revised manuscript.

Reviewer 3 Report

For researchers interested in this area, the manuscript at hand may be of interest. The aim of the study is clear and the researchers seem to have control of the statistical methods applied. However, they mix too many data in that are not related to chitosan supplementation (e.g. effects of cellulose, resistant starch and vitamin C) and should omit that. Moreover, their description of the reference search, reference selection and the applied statistical methods is incomplete. The tables are bulky and the data presentation would benefit from transferring the largest tables into the appendix and converting at least a part of the data into charts. The only figure in the manuscript lacks being mentioned in the text. The overall relevance of supplementation of chitosan on human health (the transferability of the animal experiments to public health) does not become clear from the paper.

In detail:

Abstract

line 16: I don’t think that the authors performed a publication bias – maybe they assessed it.

line 17: Q statistics and I2 values are incomprehensible statisticians’ expression – please use easy-to-understand terms in plain language and mention the direction of the effect (lowering or increasing).

line 22: TNF-a inhibition: in which organ? Or ex vivo?

line 26: “quality of human life” is too general in this context – maybe use “non-communicable diseases”?

Introduction

The Introduction needs an explanation why a meta-analysis of animal experiments was necessary. Primarily, one would rather be interested in the results of human intervention studies, due to the limitations of transferability of results from animal studies to the human setting.

line 39-40: It should be made clear here that food additives are just one possibility amongst several others to solve the problem.

line 42 Converting to what?

line 44: Replace “…treatment or the agricultural…” with “…treatment and the agricultural…”.

Results

Unfortunately, the authors limit the presentation of their results to indicating statistical indicators of little value providing an insight into the efficacy of chitosan on changes in LDL-C, HDL-C, TG etc. They should make clear where in their manuscript the absolute changes in circulating concentration of protein-standardized values for the organs can be found (in Table 3 under SMD) and provide the units of the values (presumably mg/dL). I am also convinced that, after standardizing for duration of treatment and dosing of chitosan, adding a Forrest plot would be a good idea to increase the comprehensibility of the manuscript.

It is completely unclear why the authors bring in here the results of the effect of other supplements such as resistant starch (RS) or cellulose, although this review is obviously limited to chitosan. No other component is mentioned in the introduction or the abstract, so the reader simply does not know why effects of RS, vit. C or cellulose are mentioned here and in Tables 3+5. My suggestion would be to omit results on any other component than chitosan.

Figure 1 (the Funnel plots) is not mentioned in the text.

line55: I would rather expect the Materials and Methods section to precede the Result section.

line 62: I’d rather start the sentence with “Besides in the study of Gallaher…”.

line 66: “%” of what? Weight of the diet? If so, please state that.

  1. 7 (weird – new line numbers): The abbreviations used here for the first time need to be explained.

line 9-10: If none of the “items” (I assume the authors mean differences) were significant, p should be >0.05.

Legend of Table 4: Which type of chitosan: water-soluble?

  1. 13, line 3-4: “…significantly effective in …” – by which means? Lowering? Increasing?

line 6: Strange to express R2 values as percentage.

Table 5: again, limit the results to chitosan only – all other supplements are beyond the scope of this review, alternatively you would have to change the header, Abstract, Introduction and the Materials and Methods sections.

“Estimate” – does this mean the standardized beta-value?

Discussion, p. 19

In general, the Discussion lacks pointing out the relevance of the investigated animal studies on the impact of chitosan supplementation on human health.

line 8: “catabolism is not the right word here.

line 9: “” in similar vein”: strange expression.

line14-15. One cannot say that LDL-C is “quickly” accumulating at artery walls – this process typically takes years if not decades.

line 18-19: This is nonsense. When chitosan is cationized, it can become a polycation (positively charged, as indicated by the ammonium group by the authors), but not a polyanion.

line 22: It is remarkable how far back the authors are going with their references – is the re no more actual research available on this topic?

line 28-29: There is some redundancy in the sentence “In this aqueous phase, it acts as an emulsifier on fat globules, and may have strong emulsifying properties.”

line 29-31: Isn’t rather that the fat becomes trapped within the chitosan gel than the other way around?

line 41: Should say “observed”.

line 55: Isn’t it either an animal OR an in vitro study?

line 84-95: do the authors really mean that chitosan is absorbed in the intestine? I doubt that.

Material and Methods

In general, the description of the methodology is incomplete and imprecise. It needs in detail to be pointed out how the PRISMA guidelines (as mentioned in line 99-100) were followed. Which points of the PRISMA protocol could be followed, which ones were not? Which rating system was used for the quality of the studies? Did two researchers independently screen the databases and do the literature review?

An exact description of the methodology of applying a “meta-ANOVA”, which is not a fixed statistical term, needs to be provided. Is it a multi-factorial repeated measured ANOVA? If so, the applied factors have to be clearly described.

line 102-103:  The authors need to make clear by which criteria the studies were filtered and need to indicate a list (or better a flow chart) to indicate on the basis of which criteria the vast majority of the primarily identified studies was excluded.

line 103: Change “human models” to “human studies”.

line105: Unclear what “items” means.

Conclusion

The authors should point out which aspects of chitosan action of bioindicators need to be pursued further. Where precisely are the gaps in knowledge that should be covered in future animal experiments? Furthermore, the elephant in the room should be addressed: how relevant are results of the animal studies for human health?

line 122: “Somewhat” is a very imprecise term that doesn’t help the reader to get an idea about how efficient chitosan is in improving risk factors of ischaemic stroke and cardiovascular health.

line 125: What does “useful” mean in this context? How can the results of this meta-analysis be useful?

Author Response

Thank you for your advice. 

We amended it ass you pointed. Please see below and the attachment.

===================================

Abstract

line 16: I don’t think that the authors performed a publication bias – maybe they assessed it.

--> The publication bias was performed using Egger’s linear regression test. And, Figure 1 (publication bias) removed.

line 17: Q statistics and I2 values are incomprehensible statisticians’ expression – please use easy-to-understand terms in plain language and mention the direction of the effect (lowering or increasing).

--> Amended it. Please see line 18

line 22: TNF-a inhibition: in which organ? Or ex vivo?

--> It was serum TNF-a. Please see line 22.

line 26: “quality of human life” is too general in this context – maybe use “non-communicable diseases”?

--> Amended it. Please see line 26.

Introduction

The Introduction needs an explanation why a meta-analysis of animal experiments was necessary. Primarily, one would rather be interested in the results of human intervention studies, due to the limitations of transferability of results from animal studies to the human setting.

--> Amended it as you commented. Please confirm lines 61-65 in page 2.

line 39-40: It should be made clear here that food additives are just one possibility amongst several others to solve the problem.

--> Corrected the sentence. Please see lines 40 - 41

line 42 Converting to what?

--> Revised it as you pointed. please confirm line 43.

line 44: Replace “…treatment or the agricultural…” with “…treatment and the agricultural…”.

--> Corrected it as you pointed. please see line 45.

Results

Unfortunately, the authors limit the presentation of their results to indicating statistical indicators of little value providing an insight into the efficacy of chitosan on changes in LDL-C, HDL-C, TG etc. They should make clear where in their manuscript the absolute changes in circulating concentration of protein-standardized values for the organs can be found (in Table 3 under SMD) and provide the units of the values (presumably mg/dL). I am also convinced that, after standardizing for duration of treatment and dosing of chitosan, adding a Forrest plot would be a good idea to increase the comprehensibility of the manuscript.

--> Adding a Forrest plot would be a good idea to increase the comprehensibility of the manuscript. However, since each data is important, we expressed using a Table.

It is completely unclear why the authors bring in here the results of the effect of other supplements such as resistant starch (RS) or cellulose, although this review is obviously limited to chitosan. No other component is mentioned in the introduction or the abstract, so the reader simply does not know why effects of RS, vit. C or cellulose are mentioned here and in Tables 3+5. My suggestion would be to omit results on any other component than chitosan.

--> The resistant starch or cellulose in Tables were one of references.

Figure 1 (the Funnel plots) is not mentioned in the text. 

--> Removed Figure 1.

line55: I would rather expect the Materials and Methods section to precede the Result section.

--> This is the unique format of this journal.

line 62: I’d rather start the sentence with “Besides in the study of Gallaher…”.

--> Corrected it as you mentioned.

line 66: “%” of what? Weight of the diet? If so, please state that.

--> It was percentage of the diet. Corrected it.

7 (weird – new line numbers): The abbreviations used here for the first time need to be explained.

--> We attached explanations for the abbreviations. Please confirm lines 7-9 in page 7.

line 9-10: If none of the “items” (I assume the authors mean differences) were significant, p should be >0.05.

--> It was a mistake. We amended it. Thank you for your comment. Please see line 12 in page 7.

Legend of Table 4: Which type of chitosan: water-soluble?

--> It is all types of chitosan.

13, line 3-4: “…significantly effective in …” – by which means? Lowering? Increasing?

--> It was lowering effect. Indicated it in line 3 of page 13.

line 6: Strange to express R2 values as percentage.

--> Deleted unclear sentence.

Table 5: again, limit the results to chitosan only – all other supplements are beyond the scope of this review, alternatively you would have to change the header, Abstract, Introduction and the Materials and Methods sections.

--> Revised Table 5 clearly.

“Estimate” – does this mean the standardized beta-value?

--> Yes it is.

Discussion, p. 19

In general, the Discussion lacks pointing out the relevance of the investigated animal studies on the impact of chitosan supplementation on human health.

line 8: “catabolism is not the right word here.

--> Erased the word “catabolism” as you mentioned.

line 9: “” in similar vein”: strange expression.

--> Amended it with clear word.

line14-15. One cannot say that LDL-C is “quickly” accumulating at artery walls – this process typically takes years if not decades.

--> Corrected it as you commented.

line 18-19: This is nonsense. When chitosan is cationized, it can become a polycation (positively charged, as indicated by the ammonium group by the authors), but not a polyanion.

--> It was mistake, revised it as you pointed.

line 22: It is remarkable how far back the authors are going with their references – is the re no more actual research available on this topic?  

--> Replaced references.

line 28-29: There is some redundancy in the sentence “In this aqueous phase, it acts as an emulsifier on fat globules, and may have strong emulsifying properties.”

--> Removed redundance part.

line 29-31: Isn’t rather that the fat becomes trapped within the chitosan gel than the other way around?

--> Amended it. It was a mistake. Please see line 30 in page 20

line 41: Should say “observed”.

--> Corrected it as you commented.

line 55: Isn’t it either an animal OR an in vitro study?

--> Corrected it as you commented.

line 84-95: do the authors really mean that chitosan is absorbed in the intestine? I doubt that.

--> It was a wrong typing. We revised it as you pointed.

Material and Methods

In general, the description of the methodology is incomplete and imprecise. It needs in detail to be pointed out how the PRISMA guidelines (as mentioned in line 99-100) were followed. Which points of the PRISMA protocol could be followed, which ones were not? Which rating system was used for the quality of the studies? Did two researchers independently screen the databases and do the literature review?

--> Removed unclear part.

An exact description of the methodology of applying a “meta-ANOVA”, which is not a fixed statistical term, needs to be provided. Is it a multi-factorial repeated measured ANOVA? If so, the applied factors have to be clearly described.

--> We described regarding it in lines 124-126 in page 22

line 102-103:  The authors need to make clear by which criteria the studies were filtered and need to indicate a list (or better a flow chart) to indicate on the basis of which criteria the vast majority of the primarily identified studies was excluded.

--> We judged that it was simpler to express in sentences instead of flow charts. Please see lines 109-110.

line 103: Change “human models” to “human studies”.

--> Revised it as you commented.

line105: Unclear what “items” means.

--> Attached examples for the items. Please see line 112.

Conclusion

The authors should point out which aspects of chitosan action of bioindicators need to be pursued further. Where precisely are the gaps in knowledge that should be covered in future animal experiments? Furthermore, the elephant in the room should be addressed: how relevant are results of the animal studies for human health?

--> The conclusion was described again for simplicity and clarity.

line 122: “Somewhat” is a very imprecise term that doesn’t help the reader to get an idea about how efficient chitosan is in improving risk factors of ischaemic stroke and cardiovascular health.

--> Removed it as you pointed.

line 125: What does “useful” mean in this context? How can the results of this meta-analysis be useful?
--> Corrected it as you commented.

Round 2

Reviewer 2 Report

The manuscript was signicantly improved.